# Epigenetic Reprogramming and Inheritance of the Cellular Differentiation Status Following Transient Expression of a Nonfunctional Dominant-Negative Retinoblastoma Mutant in Murine Mesenchymal Stem Cells

**DOI:** 10.3390/ijms251910678

**Published:** 2024-10-03

**Authors:** Mikhail Baryshev, Irina Maksimova, Ilona Sasoveca

**Affiliations:** Institute of Microbiology and Virology, Riga Stradins University, Ratsupites 5, LV-1067 Riga, Latvia; irina.maksimova@rsu.lv (I.M.); ilona.sasoveca@rsu.lv (I.S.)

**Keywords:** Rb1, MSCs, *Cebpa*, epigenetic reprogramming, DNMTs, adipogenesis

## Abstract

The retinoblastoma gene product (Rb1), a master regulator of the cell cycle, plays a prominent role in cell differentiation. Previously, by analyzing the differentiation of cells transiently overexpressing the ΔS/N DN Rb1 mutant, we demonstrated that these cells fail to differentiate into mature adipocytes and that they constitutively silence *Pparγ2* through CpG methylation. Here, we demonstrate that the consequences of the transient expression of ΔS/N DN Rb1 are accompanied by the retention of *Cebpa* promoter methylation near the TSS under adipogenic differentiation, thereby preventing its expression. The CGIs of the promoters of the *Rb1*, *Ezh2*, *Mll4*, *Utx*, and *Tet2* genes, which are essential for adipogenic differentiation, have an unmethylated status regardless of the cell differentiation state. Moreover, Dnmt3a, a de novo DNA methyltransferase, is overexpressed in undifferentiated ΔS/N cells compared with wild-type cells and, in addition to Dnmt1, Dnmt3a is significantly upregulated by adipogenic stimuli in both wild-type and ΔS/N cells. Notably, the chromatin modifier Ezh2, which is also involved in epigenetic reprogramming, is highly induced in ΔS/N cells. Overall, we demonstrate that two major genes, *Pparγ2* and *Cebpa*, which are responsible for terminal adipocyte differentiation, are selectively epigenetically reprogrammed to constitutively silent states. We hypothesize that the activation of Dnmt3a, Rb1, and Ezh2 observed in ΔS/N cells may be a consequence of a stress response caused by the accumulation and malfunctioning of Rb1-interacting complexes for the epigenetic reprogramming of *Pparγ2/Cebpa* and prevention of adipogenesis in an inappropriate cellular context. The failure of ΔS/N cells to differentiate and express *Pparγ2* and *Cebpa* in culture following the expression of the DN Rb1 mutant may indicate the creation of epigenetic memory for new reprogrammed epigenetic states of genes.

## 1. Introduction

Rb1 is a transcriptional corepressor 1, and one of its primary functions is to regulate the cell cycle, preventing uncontrolled cell proliferation [1,2,3]. Rb1’s function is modulated by phosphorylation mediated by cyclin-dependent kinases and can be achieved through the use of a combination of phosphoacceptor sites, cyclin/cdk docking sites, and different cyclin/cdk complexes [4]. pRb1 can exist in only the following three states: unphosphorylated, monophosphorylated, which is considered “hypophosphorylated” active pRb, and hyperphosphorylated inactive pRb, and each phosphorylation-dependent state has a unique cellular function [5]. When pRb is in its active hypophosphorylated form, it inhibits the progression of the cell cycle from the G1 phase to the S phase by binding to and inhibiting E2F1, 2, and 3a transcription factors that serve as activators of many proteins involved in the S phase [6]. This arrest of the cell cycle gives the cells time to begin their differentiation processes. In addition to its role in controlling the cell cycle, pRb is recognized as a relevant player in the differentiation of a wide range of cells [7,8]. Calo et al. reported that pRb plays a key role in the fate choice of differentiating common mesenchymal precursors between osteoblasts and adipocytes [9]. Although adipogenesis is orchestrated by the interaction of multiple transcription factors and epigenetic modifiers, encompassing multiple signaling pathways, two main transcription factors, peroxisome proliferator-activated receptor γ2 (*Pparγ2*) and CCAAT/enhancer-binding protein alpha (C/EBPα), are essential to complete the adipocyte differentiation process. It has been shown that the physical interaction between pRb1 and C/EBPα influences the transcriptional activity of C/EBPα [10]. At the molecular level, pRb has been shown to contain a pocket motif containing two functionally essential domains, A and B, which are sufficient for transcriptional repressor activity [11]. It has been shown that even a single amino acid substitution, Cys706Phe, in domain B of the Rb1 pocket can block Rb1 function, inhibiting repressor activity [12]. Interestingly, overexpression of *Pparγ*Δ5, a naturally occurring truncated isoform of *Pparγ*, has been shown to modify the *Pparγ*-induced transcriptional network, significantly impairing the differentiation capacity of adipocyte progenitor cells by acting as a dominant-negative splice isoform that reduces *Pparγ* activity [13].

The ΔS/N dominant-negative mutant (DN) of Rb1 used in the generation of the stably transfected 10T1/2 cell line was designed to eliminate six amino acids adjacent to the C-terminus of Rb1 (Figure 1a). This mutant form loses the ability to bind the protein that interacts with Rb1 and is predominantly in an unphosphorylated state while retaining the ability to interact nonspecifically [14]. In the presence of an exogenously expressed ΔS/N DN Rb1 mutant in the nucleoplasm, this mutant interferes predominantly with the function of endogenous Rb1 such that it is able to abolish adipocyte differentiation through epigenetic silencing of *Pparγ2*. This silencing status, as we have previously shown, becomes heritable, representing the epigenetic reprogramming event of a key regulator of adipogenesis whose promoter is resistant to CpG demethylation in differentiated ΔS/N cells and is unable to express *Pparγ2* [15]. Given that *Cebpa* and *Pparγ2* are considered key regulators of terminal adipocyte differentiation and that *Pparγ2* is constitutively suppressed by DNA methylation, we examined the epigenetic status of the *Cebpa* 5′ flanking region and mRNA expression to investigate whether *Cebpa* gene silencing also occurs in ΔS/N cells and whether gene-silent status is inherited. The methylation status of the promoters of the *Rb1*, Enhancer of zeste homolog 2 (*Ezh2*), Ubiquitously Transcribed Tetratricopeptide Repeat on chromosome X (*Utx*), Mixed-lineage leukemia 4 (*Mll4*), and Ten-eleven translocation 2 (*Tet2*) DNA dioxygenase genes associated with adipogenesis was also assessed to clarify the directional changes in the *Pparγ2* and *Cebpa* genes in ΔS/N cells.

We found that the transient presence of a ΔS/N DN Rb1 mutant is able to elicit permanent silencing, in addition to *Pparγ2*, the proximal *Cebpa* promoter, thereby suppressing two major transcription factors responsible for the terminal phase of adipocyte differentiation and not affecting the methylation status of the *Rb1*, *Ezh2*, *Utx*, *Mll4*, or *Tet2* genes involved in adipogenesis. *Pparγ2* and *Cebpa* gene silencing status is heritable, as confirmed by recultivating cells over an extended period of time after they have lost ΔS/N DN Rb expression and rechecking their promoter methylation status. This is the first experimental evidence of apparently “targeted” epigenetic reprogramming through transient expression of a dominant-negative Rb1 mutant.

## 2. Results

### 2.1. Alterations in Rb1 Expression in Cells That Temporarily Expressed ΔS/N Rb1

#### 2.1.1. ΔS/N Cells Overexpress Rb1 in the Undifferentiated State

To further study the consequences of the transient expression of a nonfunctional mutant, ΔS/N Rb1, which has a deletion of six amino acids, structurally representing an alpha helix of the B domain Rb1 pocket motif (Figure 1a,b), in 10T1/2 cells in the presence of functional endogenous Rb1, we compared the expression levels of endogenous Rb1 and the member Rb family p130 protein in differentiated and undifferentiated (D, UD) ΔS/N cells using real-time RT–qPCR analysis. The primer sequences are shown in Table 1. As shown in Figure 1d, moderate induction of Rb1 was observed in UD ΔS/N cells compared with wild-type cells. It has been shown that *Rb1* promoter activity is directly stimulated by its own gene product through the ATF-2 binding site [16]. Analysis of the ATF-2 binding site in the Rb1 promoter revealed that it contains a CpG, but this dinucleotide is not methylated in ΔS/N cells (Figure 1e,f); therefore, Rb1 transcription is not blocked. Apparently, the ability of Rb1 to autoinduce its own expression was used to avoid the insufficient background of endogenous Rb1 due to the existence of the chimeric Rb1:Rb1 mutant in coassembled complexes during transient ΔS/N Rb1 expression. Although the Rb1 level was moderately elevated in UD ΔS/N cells, in response to adipogenic stimuli, it was almost twofold lower than the Rb1 level in wild-type D cells (Figure 1d). How the overexpression status of endogenous Rb1 is maintained in ΔS/N Rb1 postexpression culture remains an open question. A thorough investigation of the altered epigenetic status of the Rb1 gene is necessary.

#### 2.1.2. Rb1 and p130 Show Weak Induction during Adipogenic Differentiation in ΔS/N Cells

Considering that RB1/Rb1 and RB2/p130 are indispensable in the second step of adipocyte differentiation and that the absence of RB1/Rb1 or RB2/P130 results in dysregulated adipose cells [17], we assessed the level of p130 expression in DN Rb1 mutant post-expressing cells. The p130 expression profiles in ΔS/N UD and D cells detected by RT–qPCR are similar to those of Rb1 (Figure 1c,d) in terms of the response to adipogenic differentiation, highlighting the failure of both to be induced in response to adipogenesis stimulation, as occurs in wild-type cells. It appears that the changes, if any, affected the expression of both genes similarly.

### 2.2. Dnmt3a Is Upregulated in Undifferentiated ΔS/N Cells

DNA methylation/hydroxylation regulatory enzymes are key modulators of gene transcription regulation, acting as repressors for genes unsuitable for a particular type of differentiation and activating those required by the demands of cellular differentiation, as evidenced by numerous studies of MSC differentiation into specific cell lineages. To evaluate changes in Dnmt1, 3a, and 3b expression at the transcriptional level, we quantified mRNA transcripts in ΔS/N cells and their normal counterparts under differentiated and undifferentiated conditions (Figure 2a). We observed significant induction of Dnmt3a in D wild-type and ΔS/N cells, which is consistent with a role for Dnmt3a in adipogenesis, but we also observed some upregulation of Dnmt3a in UD ΔS/N cells compared with wild-type cells (Figure 2a). It has been suggested that Dnmt3a-mediated DNA methylation is required for adipose tissue development because cells with heterozygous null mutations in Dnmt3a (lacking one allele) begin differentiating with lower DNA methylation and fail to experience the dynamic methylation alterations usually observed in WT cells [18]. However, ΔS/N cells, in which CpG methylation is still retained near the TSS of *Cebpa* (Figure 3b), fail to differentiate and do not express *Cebpa*. In contrast to Dnmt3a, which is strongly induced by adipogenic stimuli in these cells (Figure 2a), Dnmt1 is moderately activated in D wild-type and ΔS/N cells, confirming the important role of Dnmt3a in adipocyte differentiation. Notably, Dnmt3b, a de novo DNA methyltransferase, is more inert to induction in response to adipogenic stimuli in ΔS/N cells (Figure 2a), highlighting its less prominent role in adipogenesis than Dnmt3a. The induction of Dnmt1, 3a, and 3b in wild-type D cells is consistent with the function of these enzymes in adipogenic differentiation, but misregulation of Dnmt3a in UD ΔS/N cells and Dnmt3b in D ΔS/N cells appears to be caused by altered epigenetics of these genes or their signaling pathways through transient expression of the ΔS/N DN Rb1 mutant.

### 2.3. Ezh2 Is Highly Overexpressed in Undifferentiated ΔS/N Cells

During adipocyte differentiation, Ezh2 dynamically regulates the expression of key genes involved in this process by controlling the chromatin structure and accessibility of their promoters. Ezh2-mediated silencing of negative regulators of adipogenesis and activation of proadipogenic genes are crucial steps in promoting adipocyte differentiation. As shown in Figure 2b, Ezh2 is highly upregulated in ΔS/N cells compared with wild-type cells, and propagation of the repressive H3K27me3 mark appears to occur among target genes in UD ΔS/N cells, whereas Ezh2 expression is dramatically reduced upon stimulation of adipogenic differentiation in ΔS/N cells and does not change in D wild-type cells (Figure 2b).

### 2.4. Moderate UTX Induction Occurs in Differentiated ΔS/N Cells

The histone demethylase UTX is known to regulate brown adipocyte-to-myocyte remodeling in mature brown adipocytes. Demethylation of the repressive H3K27me3 mark at the *Prdm16* promoter by UTX leads to high Prdm16 expression. PRDM16 then recruits the DNA methyltransferase DNMT1 to the *Myod1* promoter, causing *Myod1* promoter hypermethylation and suppressing its expression [19]. In addition to its H3K27me2/3 demethylase-independent function, UTX has been shown to promote chromatin remodeling through the BRG1-containing SWI/SNF remodeling complex [20]. UTX binds directly to multiple genes that encode proteins that interact physically with pRB, including RB1 itself. Since we observed the greatest induction of Dnmt1 and moderate induction of Utx during the AD of ΔS/N cells (Figure 2b), we hypothesize that Dnmt1 may be recruited to the *Cebpa* promoter by Utx to silence the gene state, similar to the hypermethylation of the Myod1 promoter mentioned above.

### 2.5. Heavily Methylated Cebpa in Wild-Type and ΔS/N Cells Loses CpG Methylation during Adipogenesis, Maintaining an Unerased CpG Pattern near the TSS in ΔS/N Cells

Previously, we reported that cells that overexpress a nonfunctional exogenous ΔS/N variant of Rb1 that is lost during cell culture exhibit significant demethylation of CpG in the distal part of the *Pparγ2* promoter; however, the methylation of CpG at position 60 of its proximal part is still retained [15]. We assessed the methylation status of the 5′-flanking region of *Cebpa*, which contains 39 CpG sites located within a 510 bp core promoter. The fragment encompassing the proximal promoter was amplified, cloned, and sequenced from either wild-type or ΔS/N cells under D and UD conditions (Figure 3a). We found that wild-type or ΔS/N cells had a densely methylated *Cebpa* promoter, showing a trend toward greater methylation in ΔS/N cells in the UD state (Figure 3b). Interestingly, as with *Pparγ2*, the *Cebpa* promoter in ΔS/N cells was generally unmethylated in D cells but retained a group of methylated CpGs downstream of the TSS (Figure 3b). Analysis of *Cebpa* expression revealed complete abrogation of its expression at the mRNA level in D ΔS/N cells, but strong induction of *Cebpa* expression was observed in wild-type cells (Figure 3c).

Notably, Cebpb, another TF involved in adipogenesis, could not be induced in ΔS/N cells. A transient increase in Cebpb expression is one of the first events observed in the process of adipocyte differentiation. Cebpb acts directly on the *Pparγ2* promoter to activate its transcription, but the failure of *Pparγ2* to initiate transcription may result in the abrogation of Cebpb induction, or other mechanisms are responsible for the downregulation of Cebpb induction in ΔS/N cells.

#### 2.5.1. Unerased CpG Methylation near the TSS May Interfere with the Formation of the Overall Transcription Complex

We used the UCSC Genome Browser transcription factor binding search tool in mice (GRCm39/mm39) to analyze a 35 bp region containing an unerased methylated CpG located downstream of the TSS (Figure 3b). The results revealed that the Nrf1 transcription factor (TF), which is sensitive to CpG methylation, has a high binding score to the locus where unerased methylation is present (Figure 4a,b). The Nrf2-ECH homology 5-like subdomain of the acidic domain 1 of Nrf1 is known to be a major component that interacts with the basal transcription machinery, p300/CBP, to activate transcription [21]. In addition, the Cap’n’Collar (CNC) domain of Nrf1, a common domain shared by all CNC proteins and proteins in the CNC family, functions as a critical regulator of terminal differentiation in the blood [22], bone [23], and adipose tissue [24]. Thus, it cannot be ruled out that *Cebpa* transcription initiation is abolished because of an inability to assemble a general transcription factor complex due to the absence of Nrf1, which is unable to bind to the methylated site. Methylation-sensitive NRF1 was shown to bind only in the absence of DNA methylation, and binding was abolished by direct inhibition by CG methylation but not by indirect inhibition by MBD proteins [25]. Thus, we conclude that double CG methylation abolished Nrf1 binding to this site. The lack of *Cebpa* expression in D ΔS/N cells (Figure 3c) led us to speculate that *Cebpa* silencing in ΔS/N cells is highly selective.

#### 2.5.2. Unerased CpG Methylation Is Observed in the *Cebpa* Promoter Away from the TSS

Notably, some CpGs in the central region of the *Cebpa* promoter in wild-type cells presented a patterned, unerased distribution of methylated CpGs (Figure 3b). Since the discovery of the mCpG-dependent TF binding phenomenon, which was observed via large-scale analyses of gene expression profiles and DNA methylomes, it has been suggested that some TFs bind to methylated regulatory elements and activate gene expression [26,27]. CEBPα and CEBPβ were found to bind to a methylated sequence specifically [28]. Again, we analyzed the 35 bp region containing the methylated CpG using the UCSC Genome Browser transcription factor binding search tool. Having discovered a putative TF binding site of CEBPG that recognizes the same motif as Cebpb/a (Figure 5a,b), we propose that the unerased mCpG may represent a potential methylation-dependent motif for self-activation of *Cebpa*. In addition, other studies have suggested a role for members of the KLF family, including Klf5, Klf6, and Klf15, in promoting adipogenesis [29]. Ectopic expression of KLF2 in preadipocytes inhibits *Pparγ2* transcription, indicating its suppressive role in adipogenesis [30]. It remains to be explored whether these patterned CpGs are specific for CpG methylation readers that promote increased *Cebpa* expression in response to adipogenic stimuli or whether they represent an intermediate hydroxymethylation state toward full demethylation.

### 2.6. CpG Methylation at Position 60 of the Pparγ2 Promoter Still Occurs in Postexpressing DN Rb1 Mutant ΔS/N Cells

To confirm the existence of *Pparγ2* promoter methylation, we assessed the methylation status of the *Pparγ2* 5′ flanking region, which contains seven CpG sites located near the transcription start site. The *Pparγ2* promoter was slightly methylated in UD wild-type cells but maintained a densely methylated promoter in ΔS/N cells postexpressing ΔS/N nonfunctional Rb1 (Figure 6). Although the *Pparγ2* promoter showed pronounced CpG demethylation, constitutive CpG methylation was detected in almost all the plasmid clones analyzed at position 60 of the proximal *Pparγ2* promoter in the presence of differentiation stimuli (Figure 6).

### 2.7. The Promoters of the Adipogenesis-Related Genes Ezh2, Utx, Mll4, Rb1, and Tet2 Are Unmethylated Regardless of the Differentiation State

To assess the effects of the transient expression of the ΔS/N DN Rb1 mutant on promoter methylation and mRNA expression of the Ezh2, Utx, Mll4, Rb1, and Tet2 genes involved in adipogenesis, their promoter methylation and mRNA expression were assessed, with the exception of Mll4 expression. All selected genes are CGIs and are required for adipogenic differentiation. The corresponding promoter regions of the genes were amplified (Figure 3a) with bisulfite-specific primers. The primer sequences, number of CpGs analyzed, and amplified fragment sizes are given in Table 2, and a representative bisulfite sequencing chromatogram is shown in Appendix A.

The demethylation of the *Cebpa* promoter observed during the differentiation of 10T1/2 cells is mediated by the active oxygenase Tet2, which is known to be recruited to the CpG islands of *Cebpa* by CREB [31]. Extensive demethylation of *Cebpa* was detected in wild-type and ΔS/N cells, confirming that Tet 2 is expressed and functional. MLL4, a major H3K4me1/2 methyltransferase on mammalian enhancers, is required for the activation of cell type-specific enhancers during differentiation. During adipogenesis, MLL4 exhibits cell type- and differentiation stage-specific genomic binding and colocalizes with lineage-determining TFs and H3K4me1/2 on active enhancers [32]. Methylation analysis of 84 CpGs in the 696 bp promoter revealed a lack of CGI methylation of Mll4 in ΔS/N and wild-type cells regardless of their differentiation state (Appendix A). Therefore, it can be assumed that at the epigenetic level, Mll4 regulation is not subject to changes in DNA methylation, and Mll4 may function in adipogenic regulation. Since no tendency for methylation was observed among the *Ezh2*, *Utx*, *Mll4*, *Rb1*, and *Tet2* genes either before or after differentiation, the existing effect of the transiently expressed ΔS/N DN Rb1 mutant may not affect the expression of these genes, at least through the mechanism of promoter methylation.

## 3. Discussion

In this study, we describe the phenomenon of “targeted” epigenetic reprogramming resulting from the transient expression of the nonfunctional ΔS/N Rb1 mutant and confirm the maintenance of the reprogrammed epigenetic state in postexpression culture. Unlike mutations that prevent protein translation or disrupt protein structure, thereby causing loss of function, other mutations have alternative mechanisms, such as a dominant negative effect when the mutant protein directly or indirectly blocks the normal biological function of the wild-type protein [33]. They can thus cause (>50%) loss of function, even though only half of the protein is mutated [34]. This phenomenon is often observed for proteins that are able to coassemble into a complex with wild-type subunits, in which mutant subunits can interfere with assembly/function [35]. In an attempt to establish a stable cell line that would express the DN Rb1 mutant, we were able to detect the expression of exogenous DN Rb1 after clone selection, including the presence of an inserted DNA fragment encoding DN Rb1. However, when the ΔS/N cell line was passaged for one month, no insertions were detected, nor was the expression of exogenous DN Rb1 [15]. These cell lines were kept frozen for one year. After this period of time, the cells were thawed, passed twice for adaptation, and used for growth to isolate DNA and RNA for appropriate analysis. During the expression of the ΔS/N DN Rb1 mutant, until it is lost, we hypothesize that the following scenario may occur. Given that the deletion of six amino acids results in the dysfunction of exogenously expressed Rb1 as a transcriptional corepressor but renders Rb1 unphosphorylatable so that it is not inactivated by a natural mechanism, and assuming that the binding activity of the protein is retained, we hypothesize that mutant Rb1 interferes with the function of wild-type Rb1 by assembling Rb1 multiprotein complexes that have a normal, mutant Rb1 chimera in which the mutant protein can disrupt the activity of this complex and cause a disproportionate loss of function. Some proteins, such as BRCA1, have been shown to have RB-binding sites in both the C- and N-termini [36,37]. According to the mechanism of competitive binding, if one of the Rb molecules is a DN mutant and the other is functional, this leads to the formation of a dysfunctional molecular complex and does not allow for the hybrid complex to fulfill its natural role, reducing the efficiency of the processes carried out by Rb1 and the endogenous Rb1 context, resulting in a Rb1-null phenotype. Surprisingly, in the case of the ΔS/N Rb1 mutant, we observed the induction of Rb1 instead of a decrease in endogenous Rb1 (Figure 1d).

Considering that Rb interacts with more than 300 proteins associated with multiple metabolic/signaling pathways, the existence of many Rb-nonproducing complexes may be stressful for cells. The ability of Rb to autoinduce its own expression is used to induce Rb1 overproduction in ΔS/N cells, resulting in a lack of loss of function because of the existence of faulty Rb1-containing complexes. It has been shown that Rb1 promoter activity is directly stimulated by its own gene product through the ATF-2 binding site [17]. Analysis of the Rb1 promoter, as well as the Rb1 ATF-2 binding site, revealed that it does not contain CpG methylation in ΔS/N cells and that Rb1 transcription is not blocked (Figure 1e,f). Since Rb1 is involved in the differentiation and fate of MSCs [9], the alteration of endogenous Rb1pull caused by the formation of nonproductive complexes in which Rb1 participates dramatically affects the fate of the cell. Our results revealed that in the context of an insufficient background of functional Rb1, the expression of two major adipogenic factors, *Cebpa* and *Pparγ2*, is suppressed by CpG methylation (Figure 3b and Figure 6), whereas the methylation of the *Rb1*, *Ezh2*, *Mll4*, *Utx*, and *Tet2* genes, which are required for adipocyte differentiation, does not occur (Appendix A). The mRNA expression of Ezh2, Utx, and Rb1 was detected to a certain degree in wild-type and ΔS/N cells (Figure 1d and Figure 2b), indicating that these epigenetic regulators are not targets for epigenetic silencing in DN Rb1 ΔS/N cells. This finding is consistent with data from other researchers on the role of Rb1 in cell differentiation and fate [9].

The formation of H3K27me3 by EZH2 has been shown to be a powerful repressive mechanism to silence HIV-1 [38]. EZH2 was also shown to interact with DNMT1, DNMT3A, and DNMT3B in cancer cells and to result in the hypermethylation of genes, leading to increased permanent silencing of target genes [39]. The recruitment of DNMT3A to the 5′LTR CpG of HIV-1 upon the inhibition of UTX/JMJD3 by GSK-J4 likely results in the induction of DNA methylation. Demethylation of the repressive mark H3K27me3 on the Prdm16 promoter by UTX has been shown to result in high *Prdm16* expression. PRDM16 then recruits the DNA methyltransferase DNMT1 to the *Myod1* promoter, causing *Myod1* promoter hypermethylation and suppressing its expression [19]. Since we observed strong induction of Dnmt1 and a moderate increase in Utx during the adipogenic differentiation of ΔS/N cells (Figure 2a,b), we hypothesize that Dnmt1 may be recruited to the *Cebpa* promoter to prolong the gene silencing state, similar to the hypermethylation of the Myod1 promoter mentioned above. In addition to its H3K27me2/3 demethylase-independent function, UTX has been shown to promote chromatin remodeling through the BRG1-containing SWI/SNF remodeling complex. UTX binds directly to multiple genes that encode proteins that interact physically with pRB, including the RB gene network implicated in cell fate control [40]. Taken together, these findings suggest that altered the expression of Rb1, Ezh2, and Dnmt3a represents a new epigenetically reprogrammed state of genes, the combined action of altered activities that results in the permanent silencing of *Cebpa* and *Pparγ2* in ΔS/N cells. Interestingly, *CEBPA* is predicted to interact with the N-terminus of DNMT3a, reducing the accessibility of DNMT3a to DNA and thereby preventing promoter methylation at target genes, including those containing the *CEBPA* binding motif. This could be seen as a mechanism by which *CEBPA* helps avoid gene promoter methylation [41] in addition to the enzymatic Tet DNA demethylation that can be assumed to occur in AD in wild-type 10T1/2 cells based on successful cell differentiation and expression of *Cebpa* and *Pparγ2*.

Given that CpG methylation plays a repressive role through a variety of actions, from direct inhibition of TF binding and chromatin structure effects to repositioning of the nucleosome via the (CpG)3 methylation element [42], we assume that unerased methylation of the CpG + (CpG)2 element in close proximity to the TSS may also contribute to *Cebpa* silencing via nucleosome repositioning. Interestingly, based on nucleosome assembly experiments performed on various DNA substrates, researchers have shown that the long azidohexynyl group (ahyC, an analog of 5-mC) present at multiple CpG sites promotes nucleosome assembly. Furthermore, ahyC DNA nucleosomes can be efficiently repositioned by an Snf2H chromatin remodeler and display similar thermal stability [43]. RB1 uses the ATPase activity of Brm or BRG1 to change the nucleosome structure. This occurs in cooperation with histone deacetylases and/or histone demethylases to produce tight nucleosome structures and facilitate the formation of closed chromatin structures and RB1-mediated repression [44].

Our results suggest that epigenetic reprogramming occurs through the resetting of epigenetic marks such as DNA methylation patterns and presumably histone modifications and nucleosome repositioning that control gene expression, allowing cells to change their differentiation status or potential fate following stress-related events owing to transient expression of the mutant form of Rb1. By nature, large-scale epigenetic reprogramming occurs early in embryonic development and during gametogenesis, as well as artificially when various in vitro reprogramming conditions are used [45,46]. The described findings may reveal new conditions of epigenetic reprogramming caused by the temporal exposure of MSCs to DN Rb1. The unique effect of DN Rb1 overexpression in ΔS/N cells is that it selectively downregulates two major adipogenic genes through promoter methylation, alters the expression of Rb1 and the chromatin modifiers Ezh2 and Dnmt3a (Figure 1d and Figure 2a,b), and, to counteract the stress situation, the cells eliminate the inserted DNA fragment encoding the mutant form of Rb1 while maintaining the epigenetically reprogrammed state of ΔS/N cells. How the demand for cells to differentiate into adipocytes is disrupted under the influence of the ΔS/N DN Rb1 mutant in more detail and how new epigenetic memory is established in ΔS/N cells are questions that remain to be answered. Importantly, the exact consequences of a dominant-negative pRB mutant can vary depending on the specific mutations involved and the cell type in question. Furthermore, the context in which these mutations occur can also influence the overall effects on cell differentiation and proliferation.

## 4. Materials and Methods

### 4.1. Cell Culture

Mouse embryonic polypotent fibroblasts (C3H10T1/2 (10T1/2)) were obtained from the American Type Culture Collection (ATCC). The cells were expanded in Eagle’s basal medium (Thermo Fisher Scientific Cat. 21010-046, Carlsbad, CA, USA) supplemented with 10% heat-inactivated fetal bovine serum, 2 mM L-glutamine, and 50 μg/mL gentamicin in a CO_2_ incubator (5% CO_2_ and 100% humidity). The ΔS/N cell line that experienced transient expression of the DN ΔS/N Rb1 mutant was defrosted and cultured in Eagle’s basal medium (Thermo Fisher Scientific Cat. 21010-046) supplemented with 10% heat-inactivated fetal bovine serum, 2 mM L-glutamine, and 50 μg/mL gentamicin in a CO_2_ incubator (5% CO_2_ and 100% humidity). Cells were seeded at 4 × 10^4^/T25 flask, replenished with fresh medium on days 4 and 7, and grown to near confluence. Cells were passaged twice and then grown for RNA and DNA isolation.

### 4.2. Preparation and Electrophoresis of RNA in Denaturing Gels and Its Quantitative Evaluation

RNA was extracted from cultured cells with acidic water-saturated phenol. The RNA integrity was monitored, and the RNA amount was normalized by electrophoresis in a denaturing agarose gel supplemented with formaldehyde. Upon electrophoresis, RNA was visualized in the gel by luminescence of the 18S and 28S bands stained with ethidium bromide. The RNA amounts in different samples were normalized to the 28S band intensity in the corresponding lanes via the TotalLab Quant version 1 program product.

### 4.3. Real-Time PCR

Total RNA was isolated from 2 × 10^6^ cells with a GeneJET RNA kit (Thermo Fisher Scientific, Carlsbad, CA, USA). Purified RNA was treated with DNase I (Thermo Fisher Scientific, Carlsbad, CA, USA) (3 units/35 μL at 37 °C for 1 h). The reaction was stopped by the addition of 3.5 μL of 50 mM EDTA for 10 min at 65 °C. The RNA was quantified via a spectrophotometer at 260 nm. Its integrity was verified by the presence of 28S and 18S bands after electrophoresis on denaturing agarose gels with formaldehyde. For cDNA synthesis, 2 μg of RNA was mixed with 0.5 μg of oligo-dT18 primer. The volume was adjusted to 12.5 μL and incubated for 5 min. The reaction mixture was cooled, 4 μL of fivefold buffer, 1 μL (200 units) of Revert Aid reverse transcriptase (Thermo Fisher Scientific, Carlsbad, CA, USA), 2 μL of 10 mM dNTP mixture, and 0.5 μL of RNase inhibitor (Thermo Fisher Scientific, Carlsbad, CA, USA) were added, and the mixture was incubated at 42 °C for 1 h. The reaction was stopped by heating to 70 °C for 10 min. Real-time PCR was performed on an Applied Biosystems 7300 real PCR system (Thermo Fisher Scientific, Carlsbad, CA, USA). The parameters were as follows: denaturation at 95 °C for 5 min, melting at 95 °C for 15 s, annealing at 58 °C for 30 s, and synthesis at 72 °C for 20 s for 40 cycles. The reaction mixture was composed of 8 μL of a 2.5-fold mixture containing deoxynucleoside triphosphates (dNTPs), PCR buffer, MgCl_2_, Taq DNA polymerase SYBR Green I and ROX (Syntol, Moscow, Russia), 0.2 μL of forward and reverse primers (Beagle, Moscow, Russia) (Table 1), and 0.2 μL of cDNA and water to adjust the volume to 20 μL. The actin gene was used as a loading control. Relative gene expression was calculated via the formula R = 2^−ΔΔCT^. These experiments were repeated 3 times.

### 4.4. DNA Extraction and Bisulfite Treatment

Genomic DNA was obtained from 2 × 10^6^ cells by overnight cell incubation in TES buffer containing 0.1% SDS and 100 µg/mL proteinase K at 55 °C with subsequent phenol/chloroform extraction and isopropanol precipitation. Then, 2 µg of DNA in 50 µL of TE buffer was denatured for 15 min in 0.3 M NaOH at 37 °C. The denatured DNA was mixed with 550 µL of freshly prepared solutions of 10 mM hydroquinone and 3 M sodium bisulfite at pH 5.0 and incubated under mineral oil at 50 °C for 12 h. Bisulfite-treated DNA was desalted by isopropanol precipitation, desulfonated with 0.3 M NaOH for 5 min at room temperature, and precipitated with ethanol. Converted DNA was dissolved in 100 µL of water and stored at −20 °C. Three microliters of precipitated DNA were used for each PCR. DNA from peripheral white blood cells (WBCs) was used as a control for normal differentiated quiescent cells.

### 4.5. Bisulfite Sequencing Analysis

Bisulfite-treated DNA was used to amplify the promoter regions of genes involved in adipogenesis using primers specific for bisulfite-treated DNA (Table 2). The PCR was carried out via the use of 2.5 units of homemade Taq polymerase in a final volume of 50 µL and the following cycling conditions: 5 min at 95 °C, followed by 35 cycles (30 s denaturation at 95 °C, annealing for 30 s at 62 °C, and elongation at 72 °C for 1 min). The PCR products were gel-purified and cloned using a TOPO TA cloning kit (Invitrogen, Carlsbad, CA, USA). To prevent clonal amplification of sequences, the competent transformed cells were plated immediately after heat shock, excluding shaking bacteria for 1 h. The recombinant plasmids were isolated via the DM miniprep method [47] and sequenced in both directions using M13 forward and reverse primers (Invitrogen, Carlsbad, CA, USA) and an ABI BigDye Terminator Cycle Sequencing Kit v3.1 (Thermo Fisher, Waltham, MA, USA) with a Gene Amp 9700 PCR System (Thermo Fisher, Carlsbad, CA, USA). The sequences were detected with an ABI 3130XL Genetic Analyzer (Applied Biosystems, Foster City, CA, USA). Eighteen to twenty clones for each PCR product were sequenced and analyzed. Promoter methylation analysis was performed by aligning sequenced clones with the gene promoter region, where cytosine to uracil was converted in an in silico experiment. The efficiency of the conversion of cytosine to uracil in the bisulfite reaction was estimated as the ratio of cytosine in a non-CpG context to the total number of cytosines in the region. The clones with an efficiency of cytosine conversion less than 98% were omitted from the analysis.

### 4.6. Primer Design

The primer-BLAST software tool was used to design new target-specific primers for RT–PCR experiments. This tool is available at http://www.ncbi.nlm.nih.gov/tools/primer-blast (accessed on 30 August 2024). All amplicon primers were designed to encompass exon–exon boundaries to avoid genomic DNA amplification. The specificity of the amplicons and primer pairs was checked in silico using BLAST (National Center for Biotechnology Information) alignment tools, accessed on 1 February 2022.

To design primer-to-bisulfite region-specific matches for bisulfite-modified DNA, the following rule was applied: since the DNA strands are no longer complementary after bisulfite treatment, an individual primer set will amplify only one strand of the target sequence. The first primer was designed to anneal to the converted target sequence. The second primer was designed to anneal to the extension product of the first primer, not the opposite template strand. The PrimerSuite program (PrimerSuite, PrimerDimer, and PrimerPlex) was used to generate robust primers for PCR bisulfite sequencing analysis. The PrimeSuite, PrimerDimer, and PrimerPlex modules are all available online at www.primer-suite.com, www.primer-dimer.com, and www.primer-plex.com, respectively (accessed on 30 February 2024).

### 4.7. Bioinformatics

The Basic Local Alignment Search Tool (BLAST) commonly used in bioinformatics was applied to search for similarities and identify homologous sequences.

The UCSC Genome Browser transcription factor binding search tool in mice (GRCm39/mm39) https://genome.ucsc.edu was used to analyze TF binding in the local region of the *Cebpa* promoter.

### 4.8. Statistical Methods and Associated Software

The CFX96™ real-time PCR detection system-associated software (Bio-Rad CFX Manager 3.1, Informer Technologies, Inc. Los Angeles, CA, USA) and algorithm were used in this study. All experiments were performed in triplicate.

## 5. Conclusions

By identifying unerased, in response to adipogenic stimuli, CpG methylation near the TSS in the *Pparγ2* and *Cebpa* promoters, we postulate that epigenetic reprogramming through resetting of DNA methylation patterns and, presumably, histone modifications and nucleosomal repositioning of these genes allows cells to change their differentiation status or potential fate to counteract a stressful situation unsuitable for cell differentiation. The absence of CpG methylation in other genes required for adipogenesis, regardless of their differentiation status, suggests a highly selective process to sort out the genes most important for adipogenic differentiation for their subsequent permanent repression. A precise understanding of the molecular pathways underlying stress-associated changes in MSCs may shed light on the development of stress-associated conditions for the “targeted” cell fate reprogramming that we observed in ΔS/N DN mutant postexpressing cells.

## Figures and Tables

**Figure 1 ijms-25-10678-f001:**
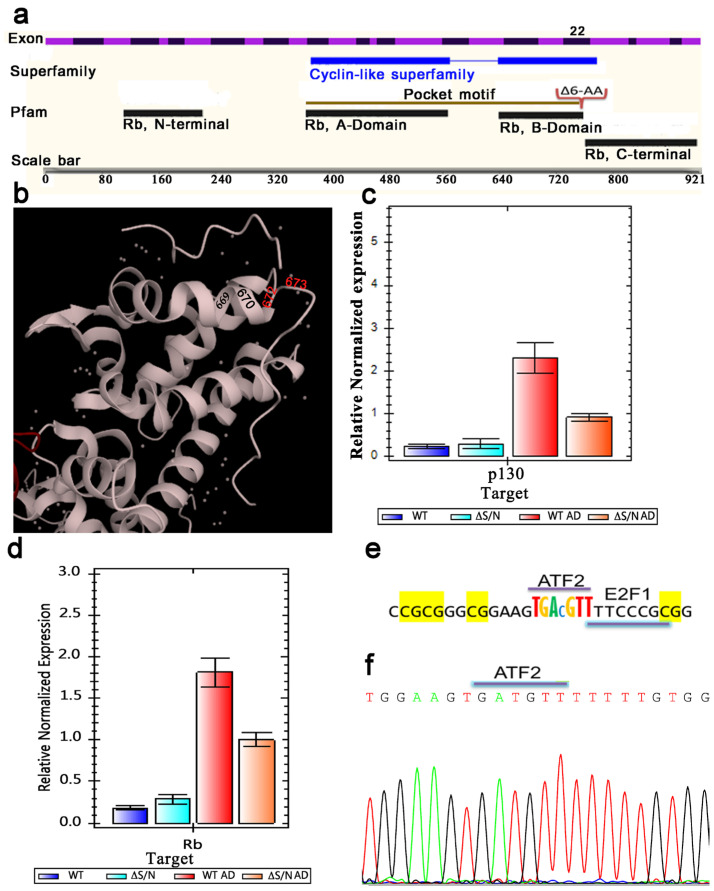
The ΔS/N cells with ILYAS amino acid deletion overexpress Rb1 in the undifferentiated state. (**a**) Schematic diagram representing the structural domains of the Rb1 protein along with the 27 exons of the gene and showing the location of the 6-amino acid deletion in exon 22. (**b**) Ribbon diagrams of the 3D atomic structure of the RB tumor suppressor bound to the transactivation domain of E2F. Deletion of the ILQYAS amino acid of the alpha helix of the RB1 B pocket domain at positions 768-763 of human RB1 resulted in 761-766 ILQYAS in mice. Adapted from 1n4m PDBe complex ID: PDB-CPX-139032. (**c**) Real-time PCR demonstrated that the p130 mRNA expression level was not changed in ΔS/N cells. (**d**) Real-time qPCR revealed that the mRNA expression level of Rb1 was greater in ΔS/N cells than in wild-type cells (n = 3); n, number of biological replicates. The error bars indicate the SEMs. (**e**) The ATF-2 binding site in the Rb1 promoter contains a CpG. (**f**) Partial chromatogram of the Rb1 promoter demonstrating that the ATF-2 binding site is not methylated.

**Figure 2 ijms-25-10678-f002:**
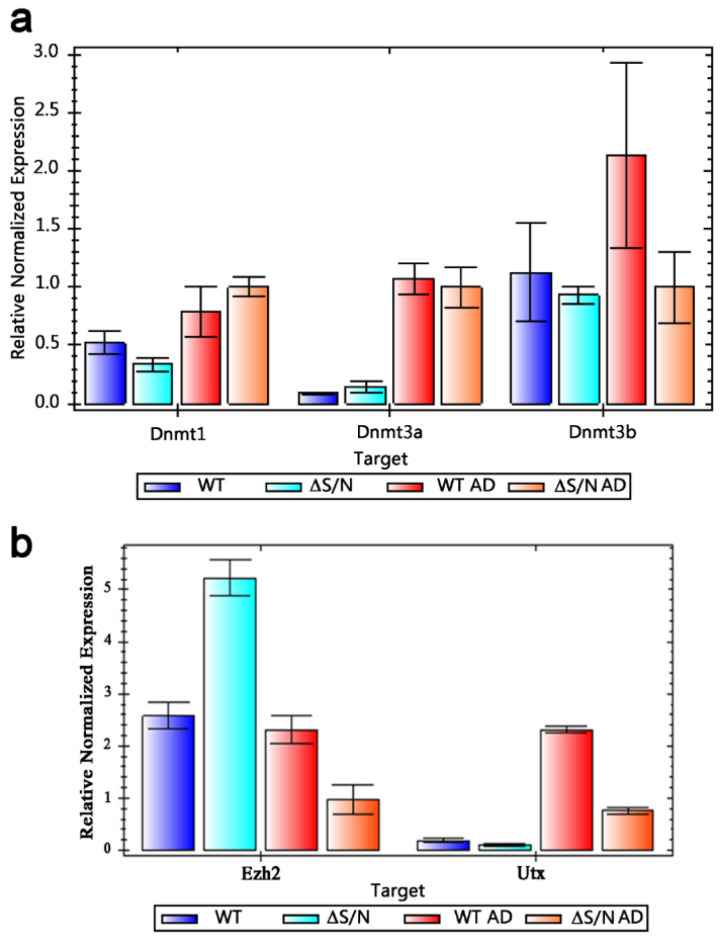
Real-time qPCR revealed that the mRNA expression levels of Dnmt3a and Ezh2 are greater in undifferentiated ΔS/N cells than in wild-type cells (n = 3); n, number of biological replicates. The error bars indicate the SEMs. (**a**) Comparison of Dnmts expression in differentiated und undifferentiated ΔS/N cells and wild-type cells. (**b**) Comparison of Ezh2 and Utx expression in differentiated und undifferentiated ΔS/N cells and wild-type cells.

**Figure 3 ijms-25-10678-f003:**
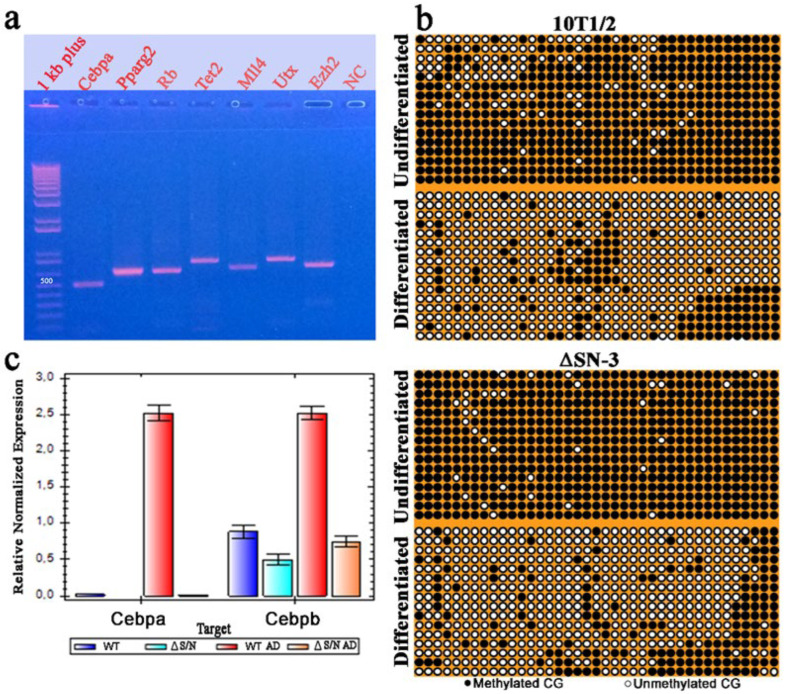
The hypermethylated promoter of *Cebpa* retained CG methylation near the TSS in differentiated ΔS/N cells. (**a**) Gel photograph showing amplification of the proximal region of the *Cebpa* gene and adipogenic-related gene promoters using bisulfite-treated DNA as a template and a pair of bisulfite-specific primers. (**b**) Bisulfite sequencing analyses of the *Cebpa* promoter. The distribution of CpG sites in differentiated and undifferentiated 10T1/2 and ΔS/N cells is shown. At the bottom of the illustrations, the methylation status of the CpGs is shown. (**c**) Real-time qPCR revealed that the mRNA expression level of *Cebpa* was lower in both undifferentiated and differentiated ΔS/N cells than in wild-type cells (n = 3); n, number of biological replicates. The error bars indicate the SEMs.

**Figure 4 ijms-25-10678-f004:**
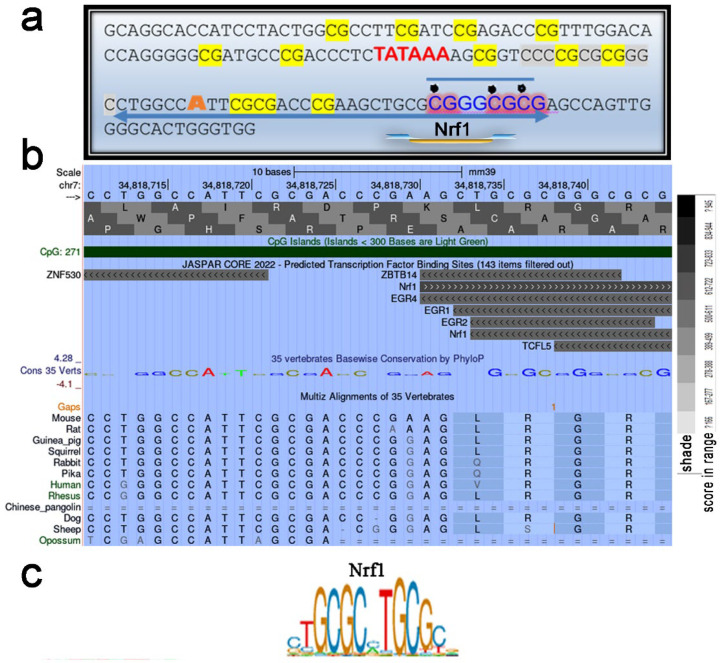
The putative Nrf1 binding site preserves the methylation of two CG sites of methylation-sensitive Nrf1. (**a**) The *Cebpa* promoter region contains the putative Nrf1 binding site. Unerased CG sites detected via *Cebpa* promoter methylation analysis upon stimulation of adipogenesis in ΔS/N cells are shown as a pictogram. (**b**) Analysis of a 35-bp region containing an unerased methylated CpG located downstream of the TSS using the UCSC Genome Browser transcription factor binding search tool in mice (GRCm39/mm39). (**c**) JASPAR NRF1 profile summary, Matrix ID MA0506.1.

**Figure 5 ijms-25-10678-f005:**
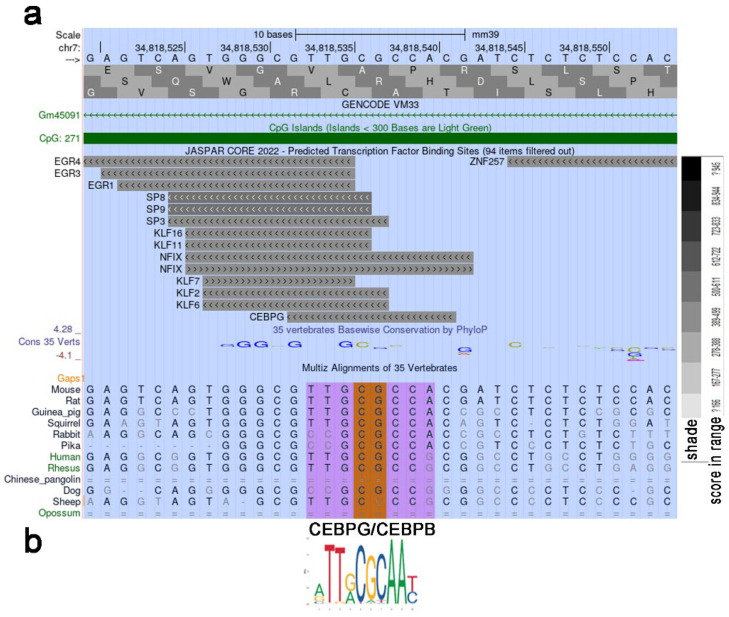
The unerased CpG methylation observed in the *Cebpa* promoter upstream of the TSS may represent a target for methylation-dependent Cebpg/b transcription factors. (**a**) Analysis of a 35-bp region containing an unerased methylated CpG located upstream of the TSS using the UCSC Genome Browser transcription factor binding search tool in mice (GRCm39/mm39). (**b**) JASPAR CEBPG/B profile summary, Matrix ID MA0838.1/MA0466.2.

**Figure 6 ijms-25-10678-f006:**
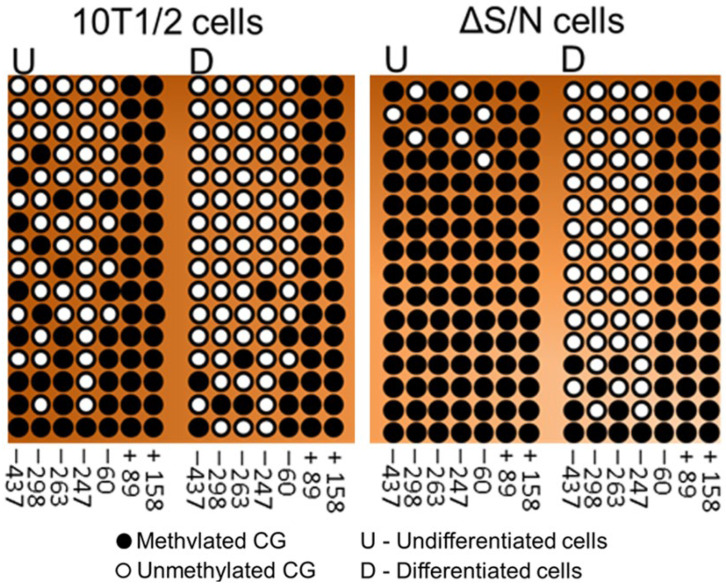
Bisulfite sequencing analyses of the *Pparγ2* promoter. The distribution of CpG sites in differentiated and undifferentiated 10T1/2 and ΔS/N cells is shown. At the bottom of the illustrations, the methylation status of the CpGs is shown.

**Table 1 ijms-25-10678-t001:** Primer sequences for RT–qPCR analysis.

Gene Symbol	Sequence (5′ → 3′) Forward/Reverse	NCBI RefSeq Locus	Tm (°C)	Size (bp)
*Cebpa*	GTAACCTTGTGCCTTGGATACT GGAAGCAGGAATCCTCCAAATA	NM_007678	60	100
Cebpb	CTTGATGCAATCCGGATCAAAC CCCGCAGGAACATCTTTAAGT	NM_009883.4	60	113
Dnmt1	GAAGGCTACCTGGCTAAAGTCAAG ACTGAAAGGGTGTCACTGTCCGA	NM_001199431.2	64	216
Dnmt3a	TGGAGAATGGCTGCTGTGTGAC CACTCATCCCGTTTCCGTTTGC	NM_001271753.2	64	223
Dnmt3b	AGTGACCAGTCCTCAGACACGA ATCAGAGCCATTCCCATCATCTAC	NM_001003961.5	64	209
Rb1	ACAGTATGCCTCCACCAGGC AATCCGTAAGGGTGAACTAGAAAAC	NM_009029	60	91
P130	TTTACTACTTCAGCAACAGCCC GAATCCCTCTCTTTTTAGTTGGAG	NM_001282000	60	93
Ezh2	ACTTACTGCTGGCACCGTCT GTTGAACAGAAAGCTGCACA	NM_007971	60	167
Utx	GGGTTGGATTATGTATTTTTAGATTT CCAACAAAAATTCTCTACCTCAAA	NM_009483	60	172

**Table 2 ijms-25-10678-t002:** Primer sequences for bisulfite sequencing analysis.

Gene Symbol	Sequence (5′ → 3′) Forward/Reverse	Chromosomal Location	Tm (°C)	Size (bp)	No. of CpGs
Rb	GAAGGTTATTAATGGTTTTATTTTGG CTCCTCACCTAACCAAAAACAA	Chr14:73430298- 73563446	62	670	110
*Cebpa*	GGGAGATAGGTTTAGTTTTAGT CACCCAATACCCCAACTAA	Chr7:34818718-34821353	62	510	64
*Pparγ2*	TTTTAGATGTGTGATTAGGAGTTT ACAATTTCACCCACACATAAATA	Chr6:115337912-115467360	62	685	7
Ezh2	TTTGTTTATGGTTTTTTTGAGAGG CAAAACCAAACTCCAAAACAAAAAC	Chr6:47507208-47572309	62	722	89
Utx	GAAGGGATATAGTTTGGATTTTTTTA TCCTCCACTATCAAACTAAAAAAC	ChrX:18028814-18146175	62	829	67
Tet2	GTTAAAGTAAATAGAAGGTGGGTT CCTTTCTAACAAATCCTACAAAACAA	Chr3:133169438-133250882	62	812	99
Mll4	GGGTTGGATTATGTATTTTTAGATTT CCAACAAAAATTCTCTACCTCAAA	Chr15:98729550-98769085	62	696	84

## Data Availability

Data are available upon reasonable request to the corresponding author.

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
