# Peer review of "Epigenetic Reprogramming and Inheritance of the Cellular Differentiation Status Following Transient Expression of a Nonfunctional Dominant-Negative Retinoblastoma Mutant in Murine Mesenchymal Stem Cells"

_ijms, 2024, doi:10.3390/ijms251910678_

Round 1

Reviewer 1 Report

Comments and Suggestions for Authors

I would like to start by complimenting the authors for this article. The study of the molecular mechanisms that are at the basis of a tumor cell such as in this case neuroblastoma are very important for studying targeted therapies. I should however make some clarifications: 

- the introduction should be enriched with some knowledge of the line and its differentiation capabilities 

- Figure 1 present in the introduction and described in subsections 2.1.1 and 2.1.2 should be modified to avoid creating confusion and therefore separated in each reference paragraph 

- Paragraph 2.2 does not speak of a figure 2b, 3a and 3c these results must be argued - I noticed that this habit of putting more figures together and arguing them in different paragraphs is repetition. Since it is a topic that requires attention in the figures it is necessary to either divide the figures or join the smaller paragraphs 

- In all the captions that refer to a graph the value of the significance of the statistical analysis must be reported 

- In all the captions the same division of the letters present in the images must be reported 

- For the rest the results, discussions and materials are clearly presented.

Author Response

Dear Reviewers,

We thank you for taking the time to carefully read our manuscript and for the valuable comments you have provided, which helped us in improving the revised paper that we are re-submitting for review.

Please find below our detailed response to each of the comments.

Best regards,

Mikhail Baryshev

Reviewer 1

I am a little confused by reviewer 1's comment below:

I would like to start by complimenting the authors for this article. The study of the molecular mechanisms that are at the basis of a tumor cell such as in this case neuroblastoma are very important for studying targeted therapies. I should however make some clarifications:

- the introduction should be enriched with some knowledge of the line and its differentiation capabilities

Am I correct in understanding which line and its differentiation should be described in introduction? Neuroblastoma? The reason for this.

- Figure 1 present in the introduction and described in subsections 2.1.1 and 2.1.2 should be modified to avoid creating confusion and therefore separated in each reference paragraph

- Paragraph 2.2 does not speak of a figure 2b, 3a and 3c these results must be argued - I noticed that this habit of putting more figures together and arguing them in different paragraphs is repetition. Since it is a topic that requires attention in the figures it is necessary to either divide the figures or join the smaller paragraphs

On the other hand, if there is a comment from one reviewer that "Overall this is an interesting and well-organized paper", what would the changes to the figures/paragraphs in response to that comment look like? These seem to be contradictory comments. I can only follow one of them.

- In all the captions that refer to a graph the value of the significance of the statistical analysis must be reported

- In all the captions the same division of the letters present in the images must be reported

- For the rest the results, discussions and materials are clearly presented.

Unfortunately, we did not find a single word in the reviewer’s preface that would mention the topic we are studying in a scientific sense. From this point of view, in most cases it is impossible to follow the reviewer's comments. Making changes to the drawing due to subjective impression is not the right decision in the author's opinion.

Reviewer 2 Report

Comments and Suggestions for Authors

In a paper entitled: Epigenetic reprogramming and inheritance of the cellular differentiation status following transient expression of a nonfunctional DN Rb mutant in murine MSCs, Mikhail Baryshev et al. presented the results of their own research on epigenetic reprogramming in mesenchymal stem cells (MSCs) in response to stress, focusing on the role of CpG methylation near the transcription start sites (TSS) of the Pparγ2 and Cebpa genes. These genes are key regulators of adipogenesis (the process of fat cell differentiation).

This study highlights the importance of studying the molecular pathways underlying the stress response in MSCs, with potential applications in regenerative medicine and the treatment of diseases in which inappropriate differentiation is a factor.

This research may have implications for understanding how transient genetic or epigenetic modifications affect long-term cellular behavior, which is important in fields such as regenerative medicine and cancer research. 

I have a few comments:

- Figure 1. should be in the Results section - the authors themselves refer to this figure in that section. 

- In the Methods, there is no information on how the cells were passaged.

- It would be worthwhile to add information on the number of cells used to isolate the DNA.

Author Response

Reviewer 2

In a paper entitled: Epigenetic reprogramming and inheritance of the cellular differentiation status following transient expression of a nonfunctional DN Rb mutant in murine MSCs, Mikhail Baryshev et al. presented the results of their own research on epigenetic reprogramming in mesenchymal stem cells (MSCs) in response to stress, focusing on the role of CpG methylation near the transcription start sites (TSS) of the Pparγ2 and Cebpa genes. These genes are key regulators of adipogenesis (the process of fat cell differentiation).

This study highlights the importance of studying the molecular pathways underlying the stress response in MSCs, with potential applications in regenerative medicine and the treatment of diseases in which inappropriate differentiation is a factor.

This research may have implications for understanding how transient genetic or epigenetic modifications affect long-term cellular behavior, which is important in fields such as regenerative medicine and cancer research.

I have a few comments:

- Figure 1. should be in the Results section - the authors themselves refer to this figure in that section.

Since we sent a Word file that contained the main text, figure legends, and figures that followed in the order listed above, the distribution of figures was not done by us. The first mention of Figure 1 can be seen in the Introduction.

- In the Methods, there is no information on how the cells were passaged.

We have added missing information in the Method section, highlighted in red.

- It would be worthwhile to add information on the number of cells used to isolate the DNA.

We have added missing information in the Method section, highlighted in red.

We replaced tables 1 and 2 due to formatting inconsistencies.

Reviewer 3 Report

Comments and Suggestions for Authors

In this manuscript, the transient expression of a ΔS/N DN Rb1 mutant prevented the expression of Pparγ2 and Cebpa through CpG methylation, while not affecting the methylation status of other genes, such as Rb1, Ezh2, Mll4, Utx and Tet2, involved in adipogenesis. Overall, this is an interesting and well organized work. However, there are several issues that need to be addressed (see comments below). 

1.     In Figure 1, please show the full name of ΔS/N AD in the caption. In addition, what is the different between the two abbreviation, AD and Differentiation (D)? If they indicate the same thing, please choose one and ensure consistent throughout the manuscript.

2.     In Figure 4, 5, 6, the caption and figure doesn’t match. 

3.     There is no evidence for inheritance of the cellular differentiation status in the manuscript. Please show evidence or discuss more about this.

4.     Please perform a thorough proofread of the revised manuscript and improve the figure quality.

Author Response

Reviewer 3

In this manuscript, the transient expression of a ΔS/N DN Rb1 mutant prevented the expression of Pparγ2 and Cebpa through CpG methylation, while not affecting the methylation status of other genes, such as Rb1, Ezh2, Mll4, Utx and Tet2, involved in adipogenesis. Overall, this is an interesting and well organized work. However, there are several issues that need to be addressed (see comments below). 

  1. In Figure 1, please show the full name of ΔS/N AD in the caption. In addition, what is the different between the two abbreviation, AD and Differentiation (D)? If they indicate the same thing, please choose one and ensure consistent throughout the manuscript.

Thanks for valuble remarks.

We put in Figure 1 full name of ΔS/N in the caption highlighted in red. AD – adipogenic differentiation and D – differentiated indicate the same in sence and abbriviation AD was replaced with D throughout the manuscript, marked in red.

  1. In Figure 4, 5, 6, the caption and figure doesn’t match. 

Thanks for valuable comment.

This is certainly true, and we have adjusted the captions and drawings in the revised version to match each other.

  1. There is no evidence for inheritance of the cellular differentiation status in the manuscript. Please show evidence or discuss more about this.

We have added information to the Discussion section for the inheritance evidence highlighted in red.

  1. Please perform a thorough proofread of the revised manuscript and improve the figure quality.

The quality of the illustrations in the proof of the edited manuscript will be improved.

We replaced tables 1 and 2 due to formatting inconsistencies.
